# Klippel–Trenaunay Syndrome, Segmental/Focal Overgrowth Malformations: A Review

**DOI:** 10.3390/children10081421

**Published:** 2023-08-21

**Authors:** Piero Pavone, Lidia Marino, Giovanni Cacciaguerra, Alessandra Di Nora, Enrico Parano, Giuseppe Musumeci, Martino Ruggieri, Agata Polizzi, Raffaele Falsaperla

**Affiliations:** 1Section of Pediatrics and Child Neuropsychiatry, Department of Child and Experimental Medicine, University of Catania, 95100 Catania, Italy; gio.cacciaguerra@gmail.com (G.C.); alessandradinora@gmail.com (A.D.N.); mruggieri@unict.it (M.R.); 2National Council of Research, Institute for Biomedical Research and Innovation (IRIB), Unit of Catania, 95100 Catania, Italy; parano@unict.it; 3Pediatrics and Pediatric Emergency Department, University Hospital, A.O.U. “Policlinico-Vittorio Emanuele”, 95100 Catania, Italy; lmarino@unict.it (L.M.); raffaelefalsaperla@hotmail.com (R.F.); 4Department of Biomedical and Biotechnological Sciences, Human Anatomy and Histology Section, School of Medicine, University of Catania, 95100 Catania, Italy; 5Department of Educational Science, University of Catania, 95100 Catania, Italy; apolizzi@unict.it

**Keywords:** Klippel–Trenaunay syndrome, inverse Klippel–Trenaunay syndrome, capillary malformations, varicosities, limb hypertrophy

## Abstract

Klippel–Trenaunay syndrome is an uncommon, infrequent, congenital disorder characterized by a triad of capillary malformation, varicosities, and tissue and bone hypertrophy. The presence of two of these three signs is enough to obtain the diagnosis. Capillary malformations are usually present at birth, whereas venous varicosities and limb hypertrophy become more evident later. The syndrome has usually a benign course, but serious complications involving various organs, such as gastrointestinal and genitourinary organs, as well as the central nervous system, may be observed. Recently, Klippel–Trenaunay syndrome has been included in the group of PIK3CA-related overgrowth spectrum (PROS) disorders. In terms of this disorder, new results in etiopathogenesis and in modalities of treatment have been advanced. We report here a review of the recent genetic findings, the main clinical characteristics and related severe complications, differential diagnoses with a similar disorder, and the management of patients with this complex and uncommon syndrome.

## 1. Introduction

Klippel–Trenaunay syndrome (KTS) is an uncommon, complex malformation disorder presenting with a triad of capillary and venous malformations and limb hypertrophy [1]. The first description of the syndrome was reported in 1900 by the French physicians Klippel and Trenaunay [2] and with similar clinical characteristic by the English dermatologist Weber in 1907 [3]. The syndrome was previously indicated with various terms such as Kippel–Trenaunay–Weber (KTW, OMIM 149000), angio-osteohypertrophic syndrome or nevus, congenital angiodysplasy, or capillary–lymphatic venous malformation (CLVM). More recently, genetic results and new clinical observations have better defined the boundaries and the clinical consistency of the KTS. In its classical form, the disorder consists of a vascular malformation syndrome variously involving cutaneous capillaries, veins, and lymphatic anomalous development, with hyperplasia of soft tissue and bones [1,4,5,6]. These findings most often are unilateral, affecting the inferior leg or arm with the lesions extending for various centimeters in length, or more rarely involving other areas of the body. Presence of at least two of the three classic components of the syndrome is considered correct for KTS diagnosis [7]. The syndrome is uncommon, and the incidence is poorly known, with no clear ethnic or sex prevalence [6]. In general, the estimated incidence is maintained to be between 2 and 5 in 100,000 individuals and is equally distributed between both genders [1]. Berry et al. [7] reviewed 49 cases of KTS, 26 females and 23 males; in all of them, the mode of transmission was sporadic. The authors [7] advanced the hypothesis that the syndrome could be due to a somatic mutation for a factor crucial to vasculogenesis and angiogenesis in phase of embryonic growth. The etiology of KTS remains not clearly established, but new hypotheses have been advanced, suggesting a genetic involvement for this sporadic condition [6]. A disorder with some of the features of KTS but with the absence of overgrowth and shortening or hypoplastic development of the muscle mass of the affected limbs has been termed “inverse Klippel–Trenaunay syndrome” [5,7].

We followed until adult age a KFS patient previously published in 1974 as a “picture of the month” [8]; a young individual with inverse KFS who is now 12 years old [5]; and a recently diagnosed 2-year-old boy with inverse KFS. These clinical observations led us to review the recent genetics results and the features of this syndrome, including complications, differential diagnosis, management, and prospective relationship to this syndrome.

## 2. Methods

The literature review was managed by obtaining clinical trials, primary research, and reviews from online bibliographic storage sites (PubMed, Scopus, MEDLINE, Cochrane Central, and Embase) between January 2003 and January 2023. The key search derived from the medical subject heading terms were pertaining to children and “Klippel–Trenaunay syndrome”, or “Klippel–Trenaunay–Weber syndrome”, or “inverse Klippel–Trenaunay syndrome”, or “capillary malformations”, or “segmental overgrowth”, or “PIK3CA”, or “PIK3CA-related overgrowth disorders”. Genetic studies, case reports, review articles, complication-related factors, and association of KTS with other disorders were manually analyzed and included in the present reference inventory. After identical records were discharged, the main search results were counted.

## 3. Results

Among the studies regarding the clinical and genetic aspects of Klippel–Trenaunay syndrome, we examined 4 articles regarding KTS review, 16 studies referring to genetics, 20 to clinical features, and 15 to complications and treatment. KTS is an uncommon disorder presenting capillary, venous, and lymphatic system impairment to various degrees in association with tissue and bone hypertrophy. KTS has been found to be correlated with mosaic-activating variants in the *PIK3CA* gene, and therefore the syndrome has been included in the group of PIK3CA-related overgrowth syndrome (PROS) phenotypes, with or without vascular anomalies [1].

## 4. Discussion

KTS is a complex and heterogeneous clinical condition. The most frequent anomaly localizations are on the limbs. These are usually found in a single limb, and rarely in multiple districts. The severity of the component vascular distortion and the degree of overgrowth segment have a relevant prognostic effect. The syndrome has usually a benign course, but complications related to the various components of the syndrome may be the cause of vascular, hematological, neurological, or other severe systemic events, thus resulting in life-threatening conditions.

### 4.1. Genetics and Link to Limb Overgrowth Syndrome

Genetic anomalies in patients with KTS have been infrequently reported. Balanced translocation 5:11 was found by Whelan et al. [9], de novo supernumerary ring chromosome 18 by Timur et al. [10], and terminal deletion 2q37.3 by Puiu et al. [11] in KTS patients. Clinical features of these patients associated with genetic anomalies are reported in Table 1. In all the patients, with regard to the cutaneous capillary, venous manifestations, and bone hypertrophy, the involvement was localized at the limbs, and mainly at the lower limbs. Cognitive status in the patients was not or only partially involved. As the age of patients at the time of the publications of the upper mentioned studies were different, a comparison among the clinical features shown by of the three patients is not feasible. However, the adult patient reported by Timur et al. [10] with Ch.18 supernumerary ring appeared to be more seriously affected. A *GNAQ* mutation in vascular endothelial cells in a patient with phakomatosis pigmentovascularis type IIb associated with KTS features was reported by Minami et al. [12]. The angiogenic growth factor 1 (*AGGF1*) (formerly *VG5Q*) was the first gene identified in a KTS patient. This gene encodes an angiogenic factor considered to be essential in vivo for both physiological angiogenesis and pathological tumor angiogenesis. This gene was identified in 5 of 130 patients with KTS in contrast to 200 control individuals in which the mutation was not found [13,14]. The 2018 classification of the International Society for the Study of Vascular Anomalies (ISSVA) defines KTS as a syndrome with capillary and venous malformations and limb overgrowth, with or without lymphatic malformation, and includes the syndrome in the group of disorders belonging to the PIK3CA-related overgrowth spectrum (PROS) [15]. Somatic mutations in the phosphatidylinositol-4-5-bisphosphate 3 kinase, catalytic subunit (*PIK3CA*), gene have been involved as primary etiological factors in KTS [2,15,16,17]. Mutations in *PIK3CA* lead to the starting of phosphatidylinositol-3-kinase (PI3K)/protein kinase and cell hyperplasia by the dysregulation of the mTORC2 pathway [17]. According to this recent advancement, the term PROS defines a group of disorders in which the core aspects are congenital or stem from an early childhood onset of segmental/focal hyperplasia with or without cellular dysplasia in the absence of comparable disorders in the family history [16]. The ISSAVA classification for vascular anomalies associated with other related *PIK3CA* mutation disorders includes, aside from KTS, several others syndromes, such as macrocephaly; the capillary vascular malformations of the lower lips; the capillary lymphatic malformations of the head and neck; the disproportions of the face and limbs, partial or generalized overgrowth involving one or more body segments (CLAPO) syndrome; the congenital lipomatous, overgrowth, vascular malformations; and epidermal nevi (CLOVES) syndrome [15,16,17].

### 4.2. Clinical Presentation

The cutaneous vascular malformations are usually present at birth, whereas venous varicosities and limb hypertrophy are seen subsequently with an average age of presentation at 4–6 years. The clinical involvement is usually unilateral and mostly affects one of the limbs. The arms are less frequently affected, and bilateral involvement has been uncommonly reported. Notable deficient growth of the affected limb may be also noted (“inverse KTS”) [5,18].

### 4.3. Inverse Klippel–Trenaunay Syndrome

Inverse KTS is a rare disorder in which the capillary and venous malformations are associated in contrast with classical KTS, with hypotrophy or shortening of the affected limb. The hypotrophy may involve bones, muscles, or subcutaneous tissues [5,18,19]. Danarti et al. [20] collected 14 cases from the literature and proposed the term “inverse KTS”. This group of 14 patients showed bone and/or soft tissue hypotrophy in association with the shortening or thinning of the involved limb, and in some cases, muscular hypotrophy was also found. Cappuccio and Brunetti-Pierri [21] reported a two-year, seven-month-old boy with vascular anomalies and localized soft tissue hypotrophy without the involvement of the underlying muscle. Ruggieri et al. [5] reported on a 3-year-old boy presenting with vascular distortion of the nevus flammeus type, continuing from the right buttock to the sole of the right foot, with leg varicosities and a primary deficiency of the soft tissues and bone. The child presented, besides macrodactyly of the first, second, and third toes, with tiny nails and cutaneous syndactyly of the second and third toes of the right foot. A brain MRI revealed exalted signal lesions in the peritrigonal areas with a regular spinal profile.

We report here a photo of a new case of a 24-month-old boy recently admitted to this institution in which a diagnosis of inverse KTS was made. The child presented with capillary, venous anomalies, and bone and tissue hypogrowth, affecting the inferior right limb. In the child, a big bilateral toe was also present (Figure 1 and Figure 2).

### 4.4. KTS Malformations

In classical KTS, a cutaneous birthmark, the “port-wine stains”, is the first clinical sign, and it is usually present at birth, initially with a patchy distribution. The capillary malformations consist of abnormal ecstatic capillaries in the papillary dermis, presenting with thin capillary walls. The lesion appears flat, being bluish to purplish in color, and it is present in almost all cases. It tends to increase as the child grows, and the color may become lighter or darker [22,23,24].

Venous malformations are a frequent manifestation of KTS, presenting with varicosities involving both superficial and deep venous systems with variable expression ranging from ectasia of small veins to persistent embryonic blood vessel and huge venous deformity of the superficial venous system [2,22,25]. The persistent embryonic veins may be the cause of the reduced blood flow, with subsequent deep-vein thrombosis and associated pulmonary embolism [25,26,27].

Lymphatic malformations are reported in around to 15% of the cases and are represented by dilated vessels, filled with clear proteinaceous fluid, usually not connected with normal lymphatic vessels. Circumscribe microcystic and/or macrocystic lymphatic malformations of the affected body segment may also occur [14,24].

Regional hyperplasia may involve soft tissue, muscle, and bone of the affected region. The severe limb hypertrophy may cause limb-length discrepancy and problems with walking [2].

Congenital malformations include, among others, syndactyly, macrodactyly, and polydactyly, and they have been reported in about 29% of the affected subjects [5,27,28,29].

The single involvement of the triad KTS component that is capillary, venous malformations and bone hypertrophy is variously distributed according to the results of the experience reported by the authors on this topic (Table 2). Servelle et al. [30] among 786 patients with KTS, found elongation of the impaired limb constantly present in all patients, edema in 84% of cases, varicose veins in 36%, and flat angiomatosis in 32%. Venography and surgical investigation showed deformity of the deep veins, affecting the popliteal vein in 51%, superficial femoral vein in 16%, both popliteal and superficial femoral veins in 29%, iliac veins in 3%, and lower vena cava in 1%. Gloviczki et al. [31] reported on 144 patients (65 male and 79 female). Among these, hemangioma was found in 137 patients (95.1%), varicosity in 110 (76.4%), and hypertrophy of the soft tissues or bones in 134 (93.1%). In a wide number of the patients, the distortion involved only one lower extremity. Most of the patients did well without medication or only with elastic compression. Jacob et al. [32] reported on a series of 252 patients with KTS, with 136 females and 116 males. Among these, capillary malformations (port-wine stains) were present in 246 patients (98%), varicosities or venous malformations in 182 (72%), and limb hypertrophy in 170 (67%). All the three elements of KTS were found in 159 patients (63%), and 93 (37%) showed two of the three classical aspects. Atypical blood vessel, including lateral veins and persisting sciatic vein, resulted in 182 patients (72%). The frequency of various congenital vascular malformations in KTS was reported by Yamaki et al. [26]. In this study, sixty-one patients with KTS were enrolled. Among these patients, 45 (74%) had mainly venous defects, 4 (6%) had mainly lymphatic deficiency, and 12 (20%) had mixed vascular defects. Capillary malformations were reported in 54 patients (89%), among which port-wine stains were the most prevalent (40 patients, 66%), telangiectasia was reported in 31 patients (51%), and angiokeratoma in 18 patients (30%). Extratruncular venous malformations were reported in 47 patients (77%), and truncular venous deformities were reported in 50 patients (82%). In the group of 50 patients, embryonic lateral marginal vein was frequently reported (32 patients, 53%). The authors [26] maintained that reflux in the embryonic vein was found in only nine patients (15%). Twelve patients (20%) had reflux in the great saphenous vein, and four (7%) had reflux in the small saphenous vein. Seven patients (12%) showed deep-vein hypoplasia, and only five patients (8%) showed deep-vein aplasia. In 13 patients (21%), extratruncular lymphatic distortions were found, and truncular lymphatic malformations were reported in 17 patients (28%). Alwalid et al. [23] reported on the results of the imaging features found in 14 patients with KTS: unilateral lower limb involvement in 10 patients (71%), bilateral, asymmetric lower limb involvement in 4 (29%), varicosities in 13 (93%), muscle hypertrophy in 11 (79%), and venous anomalies in 9 (64%). Less common anomalies consisted of lymphedema in four patients (29%), arterial malformations in four (29%), soft tissue hemangioma in three (21%), pelvic and thigh phleboliths in three (21%), venous aneurisms in three (21%), bone abnormalities in two (14%), and lymphadenopathy in two (14%).

### 4.5. Clinical Complications

Clinical complications affecting various organs have been reported in individuals with KTS.

The gastrointestinal system involvement may cause abdominal pain, bleeding from mild to severe, usually linked to venous malformations and varicosities. Diffuse cavernous hemangiomas of the distal colon and rectum are one of the most common events of gastrointestinal bleeding and have been estimated to occur in between 1% and 12.5% of KTS patients [33,34,35,36,37,38,39].

The genitourinary system has been also involved, even if not frequently. Rubenwolf et al. [40] maintained that the urinary tract in KTS is implicated in about 10% of the cases. These authors [40] reported a patient with an extensive lympho-venous distortion to the bladder, which led to massive, recurrent gross hematuria. Furness et al. [41] showed that in KTS, the overall genito-urinary symptoms occur mainly in patients with the most severe types and generally involve the cutaneous vascular malformations of the trunk, pelvis, and genitals. The authors reported two KTS patients in which intra-abdominal vascular and intrapelvic extension of the vascular malformation was associated with the lower abdominal pelvic cutaneous implication of the genitalia [41]. In cases of KTS, the vascular malformations of the bladder were the cause of massive hematuria [42], and they also required nephrectomy [43]. Lower gastrointestinal bleeding, hematuria, and splenic hemangiomas were reported by Kocaman et al. [44]. Hematuria concurrent with rectal bleeding has been also reported [42].

Pulmonary complications: Venous thromboembolism is a severe pulmonary complication in KTS patients, which may be followed by pulmonary hypertension and by right heart failure [37,45,46]. Recurrent pulmonary embolism related to deep venous thrombosis may be followed by chronic changes in the small vessels, leading to pulmonary hypertension [2].

Neurological complications are uncommon but severe. Brunaud et al. [47] reported on two patients presenting with KTS and neurological complications: the first showed flaccid paraplegia and occlusion of the anterior spinal artery at the spinal cord angiography of a probable thrombotic origin; the second patient complained of left hemiplegia caused by a right superficial sylvian artery infarct. Ischemic stroke secondary to a thromboembolic event was reported by Lee and Choi in a 43-year-old man [48]. Multiple giant intracranial aneurysms [49], intracranial arteriovenous malformation [50], and cerebral and spinal cavernomas [51] have been reported as examples of neurological manifestations in patients with complicated neurological manifestations of KTS. The involvement of the peripheral nervous system was found in a 67-year-old woman with KTS. In the patient, microscopic analysis showed the existence of epineural arteriovenous anastomoses and endoneurial vascular coils in a sural nerve biopsy from both hypertrophic and non-hypertrophic limbs [52].

In KTS patients, various types of cancers have been reported. In children, the risk of embryonal cancer other than Wilms tumor does not appear to be higher compared to people of the same age [53]. Recently a KTS child with chronic myeloid leukemia has been reported [54].

Chronic pain is a common problem in this syndrome and may have various origins, including cutis infections, thrombophlebitis, gangrene, lymphedema, and other causes that underly clinical events.

### 4.6. Clinical Evaluation

In the classical type of KTS with lesions localized in the limb, the diagnosis may be reached clinically. X-ray of the affected limb, as well as ultrasound, CT, MRI, and color Doppler of arteries and blood vessels, are useful to confirm the presence of the various venous anomalies involved in patients. Genetic analysis may be a further factor to confirm the diagnosis. Thoracic X-ray, as well as serial laboratory findings including D dimer assay, red blood cell count, fecal occult blood test, and urinalysis, are appropriate exams to reveal red blood cell deficiency, as well as thorax and genitourinary problems. MR angiography and catheter angiography may be useful to indicate the severity of the complications. However, invasive exams should be performed with particular caution, as they may be the cause of vascular complications.

### 4.7. Differential Diagnosis

Proteus syndrome (PS) is a disorder characterized by unbalanced and disproportionate hyperplasia, connective tissue nevi, epidermal nevi, dysregulate adipose structure, and vascular malformations [5]. Individuals with PS may show signs that occur also in patients with inverse KTS, such as segmental (patchy) areas of inadequate growth, enumeration of the skin and adipose tissue, and impacts on small or larger regions of the body. These phenotypes are known as elattoproteus. A mixture of proliferation and hypogrowth may be seen in patients with PS [5]. In classical clinical presentation, PS and KTS may be easily distinguished, and results of the genetic analysis may be diriment as PS patients may show positivity of single somatic activating AKT1 c. 49 G>A p. E17K variants [55].

Parkes Weber syndrome (PWS) is the condition most similar to KTS. PWS is characterized by limb hypertrophy, ipsilateral cutaneous hemangioma, and varicosities with arterio-venous fistula. PWS has features of KTS, but deep malformations in PWS are arteriovenous, whereas in KTS, they are venous. Genetic analysis provides different results: *PIK3CA* mutation is indicative for KTS and *RASAI* for PWS [2,56].

CLOVES syndrome (OMIN number 6129918) is characterized by congenital lipomatous overgrowth, vascular malformations, epidermal nevi, and scoliosis/skeletal/spinal anomalies. The condition is related to somatic mutations in *PIK3CA* [57]. Clinical differentiation between CLOVES and KTS consists of the absence of truncal involvement and the preferential location of the anomalies in limbs in KTS patients [54]. Moreover, high-flow atrio-ventricular malformations (AVM) and spinal/paraspinal AVM are features of CLOVES and PWS but have not been reported in patients with KTS [58].

### 4.8. Management

At present, there is no curative treatment for patients with KTS, and treatment is prevalently conservative and symptomatic, mainly directed towards enhancing the patient’s quality of life. This is particularly the case for children with this disorder, in which the course is usually benign. Compressive therapy bandages or elastic garments are wrapped around the affected limbs to help prevent swelling, problems with varicose veins, and ulcers. Elastic stocking, limb elevation, and intermittent pneumatic compression devices associated with physiotherapy have been shown to have beneficial effects [33]. This may be a useful psychotherapeutic support. Particular attention should be dedicated to the epidermis to prevent superficial infections and bleeding due to scratching. Sclerotherapy for capillary venous and lymphatic malformations has been employed with good outcomes. In intensive port-wine stain treatment, laser therapy may be indicated. Orthopedic intervention has been suggested in the case of a length variation projected to exceed 2.0 cm at skeletal maturation, which can be managed with epiphysiodesis in the growing child [32]. Other indications for surgery include complication events causing hemorrhage, infections, and venous thromboembolism [31]. Rapamycin has been used to reduce the progression of the vascular malformation. The drug has an effect of blocking the P13K/AKT/mTOR pathway. The rapamycin–protein complex inhibits the action of mTOR1, inducing the arrest of cell growth, avoiding tissue proliferation. Sirolimus has been employed in twenty-nine patients with capillary lymphatic venous malformations including KTS and CLOVES patients. Ninety-three percent of patients reported improved quality of life, and 86% had an improvement in at least one of their symptoms, Side effects included neutropenia, lymphopenia, infections, and aphtous ulcers/stomatitis, but no signs of toxicities [59].

### 4.9. Prognosis

The course of the disorder is usually good, particularly in younger age groups, but may become severe, depending on the underlying conditions associated with the syndrome. Several factors may predispose to thrombosis in patients with KTS, and venous malformation may be the cause of intravascular coagulopathy with subsequent thrombosis and thromboembolism. Bleeding from the gastrointestinal tract vascular malformations should be carefully followed, as it may be occult and chronic, causing severe anemia.

### 4.10. Present and Future Perspective

KTS is a very complex disorder with a wide variety of clinical expressions. Clinical features defining KTS have been established and consist of the presence of the classical triad of capillary and venous malformations and limb hypertrophy. Diagnosis may be performed also in the presence of two of the three classical signs. The etiological basis of this disorder is not completely clear and is limited by several factors such as the uncommon frequency of the disorder; sporadic condition, even if familial cases and intrafamilial symptoms have been reported; variability of the expression of the clinical features and of the complications linked to the underlying condition; and difficulty in the surgical treatment of the clinical complications of the implicated organs, including lung, gastrointestinal tract, liver, and kidney. Various pathogenetic hypotheses have been proposed, including chronic venous hypertension, persistence of the embryological vascular system, and generalized developmental mesodermal anomaly [33]. A new road has been opened for the etiological basis of this syndrome on the evidence that the more severe cases of KTS have been linked to mosaic-activating variants in the *PIK3CA* gene. On this basis, KTS has been recognized not as a distinct clinical entity but as part of the wide group of the PIK3Ca-related overgrowth spectrum (PROS). It has been hypothesized that atresia or obstruction of the dep veins in the leg as a first event may lead to chronic venous hypertension with subsequent anomalous development of nevus, varices, and hypertrophy [30,31,32,33]. The inclusion of the KTS within the PROS group may be useful to reach new advances in the etiopathogenetic factors of the KTS.

In patients with KTS, further research should address drug treatment with the aim of preventing the progression of the anomalies that are the cause of severe clinical complications. Some positive results came from the use of rapamycin. Sirolimus, also known as rapamycin, is an allosteric inhibitor of mTOR, acting by inhibiting the P13K/AKT/mTOR pathway. It has been used in some medical disorders, especially as an immunosuppressive medication to avoid organ rejection, as an anti-angiogenic treatment on coated cardiologic stents, and as a cytostatic agent in breast and renal cancer, and it has been shown to have efficacy in selected vascular malformations [60]. Rapamycin has been used in KTS to block the advancement of vascular malformations and of cellular growth, thus improving the quality of life of patients with KTS. As rapamycin has been shown to have some adverse effects, its use in KTS deserves further control. Studies on this topic are extremely useful in the treatment of patients with KTS.

### 4.11. Limitations

This review of patients affected by KTS shows several limitations mainly related to the heterogeneity of the clinical manifestations and the difficulty in finding a common cause of this complex disorder. Moreover, the disorder shows a very different evaluation according in terms of age, since it is usually benign in childhood and becomes more severe with further complications related to the disorder. In the present study, we tried to collect the main clinical characteristic of this complex disorder.

## 5. Conclusions

KTS is a disorder that may be clinically banal, but which may cause severe functional and aesthetic problems and also is associated with severe complications. There is no uniformity in the clinical presentation of the triad that constitutes the syndrome, in extension of the anomalies and the complications linked to the disorder. Treatment of complications are often dangerous and difficult to solve. Many advances have been obtained by new genetic research, which has validated that a mutation in the *PIK3CA* gene has been involved in KTS and members of the related limb overgrowth spectrum. Therapeutic treatment with rapamycin should be extensively employed with the hope to improve the clinical condition of the patients and to prevent complications. Adverse effects caused by the drugs used for the treatment of patients with KTS should be carefully evaluated.

## Figures and Tables

**Figure 1 children-10-01421-f001:**
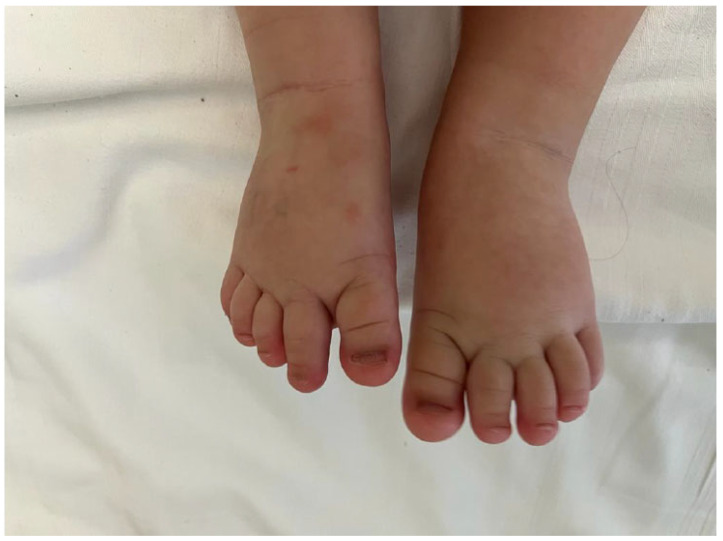
Big bilateral toe.

**Figure 2 children-10-01421-f002:**
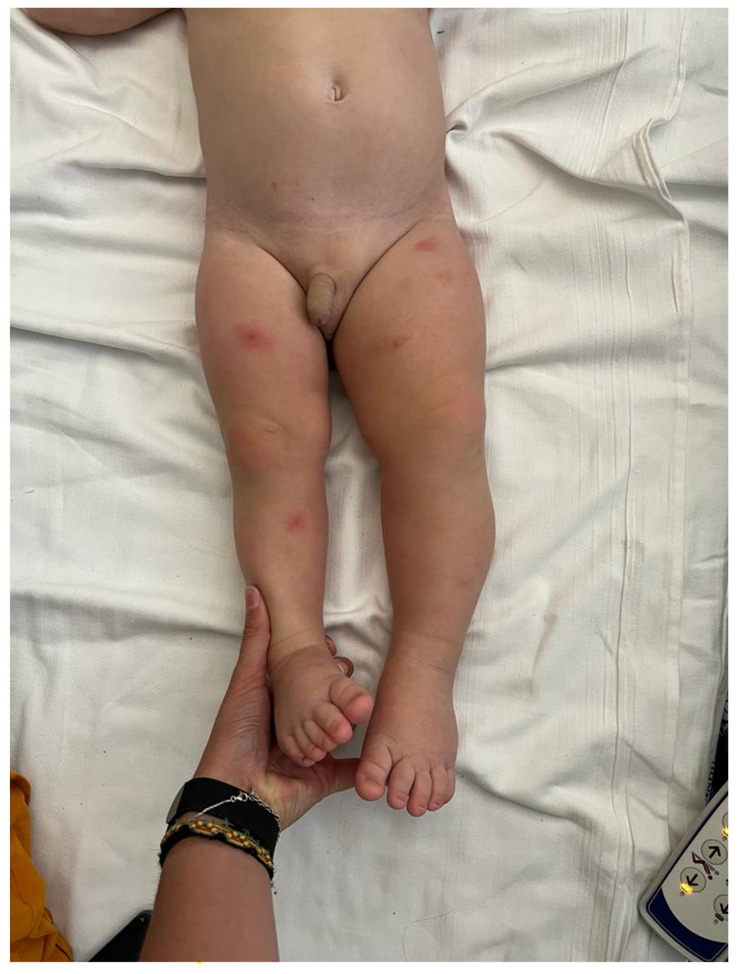
Bone and tissue hypogrowth affecting the inferior right limb.

**Table 1 children-10-01421-t001:** Clinical Features of KTS patients and chromosomal anomalies.

	5:11 Balanced Translocation Whelan et al. [9]	2q37.3 Terminal Deletion Puiu et al. [11]	Ch18 Supernumerary Ring Timur et al. [10]
Age	from Birth	10 years	30 years
Gender	Female	Male	Male
Capillary, venous, lymphatic malformations	Right arm capillary hemangioma, left trunk vascular anomaly extending to left thigh	Right lower limb, trunk, and upper limbs hemangiomatosis	Several “bumps,” and varicose veins on right lower limb.
Hypertrophy	Right left leg (first noted in the 5 years of life)	Upper limbs, scapular girdle, right lower limb.	Right lower limb (noted at 8 months)
Extremities	Right toe larger. Bilateral pes cavus	Toe malformations	Long tapering fingers, elongated, and thin feet
Others	Large reticulated mildly erythematous vascular lesion on left anterior thorax and abdomen extending inferiorly to left thigh, superiorly to amterior neck and right shoulder and posteriorly to the thoracolumbar spine.	Short stature, failure to thrive. Left lower limb atrophy, scoliosis	Multiple collateral vessels or varicosities in the region of adrenal glands, gallbladder and in the region of the porta hepatis. Severe vascular malformations in the rectum and colon, and consumptive coagulopathy. Splenomegaly, kidney asymmetry and perirectal calcification
Cognitive status	Normal	Moderately retarded. Febrile seizures	Learning disability

**Table 2 children-10-01421-t002:** Number and percent of the single component of the KTS triad.

Authors	No. of Patients	Capillary	Venous	Hypertrophy
Servelle et al. [30]	786	Flat angiomatosis	Varicosity	Bone hypertrophy
		251 (32%)	283 (36%)	all patients
Glovkzki et al. [31]	144	Hemangioma	Varicosity	Bone hypertrophy
		137 (95%)	110 (76%)	134 (93%)
Jacob et al. [32]	252	Capillary malformations	varicosities or venous	Limb hypertrophy
		182 (72%)	170 (67%)	246 (98%)
Yamaki et al. [26]	61	Capillary Malformations	venous, lymphatic mixed defects,	N:R
		54 (89%)	45 (74%), 4 (6%), 12 (20%)	54 (89%)
Alwalid et al. [23]	14	Lower limb	Varicosity	Hypertrophy
		10 (71%)	13 (93%)	11 (79%)

## Data Availability

The data used to sustain the findings of this study may be released upon application to the corresponding author who can be contacted at ppavone@unict.it.

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
