# Peer review of "Klippel–Trenaunay Syndrome, Segmental/Focal Overgrowth Malformations: A Review"

_children, 2023, doi:10.3390/children10081421_

Round 1
Reviewer 1 Report
This manuscript entitled: Klippel-Trenaunay syndrome, segmental/focal overgrowth malformation: a reappraisal of an uncommon disorder” explored the main genetic findings, the main clinical characteristic, related severe complications, and the management of patients with KTS syndrome.
The manuscript is well-written and intetesting, however few issues need to be edited:
1. The manuscript has method and results section so it is a systematic review. Please add “A systematic review” in the title.
2. I have noticed few typing and grammar errors such as “Cochraine” it should be “Cochrane”, Please revise all the manuscript regarding this issue.
3. The results section is very short and missing many important data. How many studies were retrieved and included in this systematic review. What was the types of the retrieved and eligible studies?. How many subjects included as the total of subjects from the included studies?. It would be better if a table describe the main characteristics of the included studies added to the manuscript.
A few grammar and typing errors need to be corrected.
Author Response
Reviewer 1
Q 1. The manuscript has method and results section so it is a systematic review. Please add “A systematic review” in the title.
R 1 We have add “A systematic review” in the title.
Q 2. I have noticed few typing and grammar errors such as “Cochraine” it should be “Cochrane”, Please revise all the manuscript regarding this issue.
R2 We have ceck this issue
Q 3. The results section is very short and missing many important data. How many studies were retrieved and included in this systematic review. What was the types of the retrieved and eligible studies?. How many subjects included as the total of subjects from the included studies?. It would be better if a table describe the main characteristics of the included studies added to the manuscript.
R 3 We have added tables
Reviewer 1
Q 1. The manuscript has method and results section so it is a systematic review. Please add “A systematic review” in the title.
R 1 We have add “A systematic review” in the title.
Q 2. I have noticed few typing and grammar errors such as “Cochraine” it should be “Cochrane”, Please revise all the manuscript regarding this issue.
R2 We have ceck this issue
Q 3. The results section is very short and missing many important data. How many studies were retrieved and included in this systematic review. What was the types of the retrieved and eligible studies?. How many subjects included as the total of subjects from the included studies?. It would be better if a table describe the main characteristics of the included studies added to the manuscript.
R 3 We have added tables
We wish to thanks the reviewer 1 for the nice suggestion
Reviewer 2 Report
In this manuscript, Pavone et al., describe a narrative review of a very uncommon congenital disorder, named KLIPPEL-TRENAUNAY-WEBER SYNDROME, which includes the main clinical and the recently described genetic features.
The manuscript is in general well structured and written, however, it is difficult to identify a major contribution to the field, despite its relation with a very rare disease. Although
data retrieved from the literature is well described, but it is not sufficiently contrasted (i.e. use of Tables to contrast clinical/genetic findings among different case series).
In my opinion, another major limitation of this review is the absolute lack of representative clinical and radiological images of the KLIPPEL-TRENAUNAY-WEBER SYNDROME, which is of utmost importance to
provide the reader with a clear clinical picture of this disorder, which even allows the differential diagnosis.
Other minor criticisms include:
The official name of this disorder: KLIPPEL-TRENAUNAY-WEBER SYNDROME(OMIM %149000).
Please review the phrase "selected from January 2003 to January 2023"
Please write the human gene symbols in italics.
Section 4.6: Please review the correctness of the phrase "there is no care for patients with KTS", is it refers to the absence of "curative treatment" or "treatment consensus"?
Author Response
Reviewer 2
- 1 The manuscript is in general well structured and written, however, it is difficult to identify a major contribution to the field, despite its relation with a very rare disease. Although
data retrieved from the literature is well described, but it is not sufficiently contrasted (i.e. use of Tables to contrast clinical/genetic findings among different case series).
- 1 Presently KFS has been included in the group of disorder belonging to the PIK3CA-related overgrowth spectrum (PROS). Case-reports with chromosomal anomalies are scanty. As suggested, we reported a table I with three examples of KFS associated with chromosomal anomalies and Table II reassuming the results of Imaging features as reported by ALWALID et al.in 14 KTS patients (see Table 1 and Table 2)Line ….. Line……
- 2 In my opinion, another major limitation of this review is the absolute lack of representative clinical and radiological images of the KLIPPEL-TRENAUNAY-WEBER SYNDROME, which is of utmost importance to
provide the reader with a clear clinical picture of this disorder, which even allows the differential diagnosis.
- 2 We have included a photo of a recent child
Minor criticism
- 3 The official name of this disorder: KLIPPEL-TRENAUNAY-WEBER SYNDROME(OMIM %149000).
- 3 The official name of the disorder has been reported. Introduction Line 4
Q.4 Please review the phrase "selected from January 2003 to January 2023"
- 4 The sentence has been corrected Methods line 3
- 5 Please write the human gene symbols in italics
R 5: Italics symbol has been used in human gene
Q.6, is it refers to the absence of "curative treatment" or "treatment consensus"?
- 6 the sentence has been corrected as you suggested Management Line 1
Q The manuscript requires language editing
R The paper was review by a full professor of English of the University of Catania
We wich to thanks reviewer 2 for the appreciated suggestion
Reviewer 3 Report
Even though this manuscript could be an interesting review, there are many points that need clarification. It is suggested to review the design in detail, present the most significant points and that the best description/review of these adhere to the main objective that led them to carry out this review.
1. The manuscript requires language editing. The title should improve, for example “Klippel-Trenaunay syndrome, segmental/focal overgrowth malformation: a review”. The main aim must be direct and the same throughout the manuscript. References should be recent and relevant, they should be well referenced, and their use should be improved throughout the manuscript.
2. The introduction section needs improvements. It would be better to write this here: (KTS). A proper presentation and a good and clear justification (reason) for conducting this comprehensive review should be given. Why was this review done?
3. The methods section is sparse and need deep improvements. What kind of review was it? The description must be clear, concise, and detailed. What parameters were included and excluded? What data was essential to collect?
4. The results section should improve: How many articles did this search return? Describe the most important characteristics of the studies found, for example, the number of patients in total, age, gender, etc. Avoid repeating the same information. Can the authors somehow represent the phenotypic and treatment characteristics in a graph, figure, or flowchart? It should be clear which were the most significant information collected from this review.
5. The discussion section should start with the main objective of this review study and the most important results found. The collected results (subsections) by the authors should be discussed from multiple angles and placed in context without overinterpreting them. What was known, what new information did the authors collect? What does this review contribute to the current knowledge about this syndrome? A paragraph of limitations and suggestions for this review should be written before the conclusion.
6. Did the authors reach any conclusions from this review?
I would like to encourage the authors to rewrite this review, thinking about the main objective of this study, its design and responding with the results and arguments of the discussion to the most appropriate conclusion of this review.

The manuscript requires language editing.
Author Response
Reviewer 3
- 1 The manuscript requires language editing. The title should improve, for example “Klippel-Trenaunay syndrome, segmental/focal overgrowth malformation: a review”. The main aim must be direct and the same throughout the manuscript. References should be recent and relevant, they should be well referenced, and their use should be improved throughout the manuscript.
- 1 The text has been reviewed by an English Native reviewer. The title of the text has been changed as suggested
Q 2 The introduction section needs improvements. It would be better to write this here: (KTS). A proper presentation and a good and clear justification (reason) for conducting this comprehensive review should be given. Why was this review done?
R. 2 KTS acronym has been reported at the beginning, The reason of this comprehensive review has been added at the end of the introduction section (Last 5 lines of the introduction 9
- 3 The methods section is sparse and need deep improvements. What kind of review was it? The description must be clear, concise, and detailed. What parameters were included and excluded? What data was essential to collect?
R: 3 Method section has been modified Methods Line 6-9
Q.4 The results section should improve: How many articles did this search return? Describe the most important characteristics of the studies found, for example, the number of patients in total, age, gender, etc. Avoid repeating the same information.
R, 4 Result section has been modified Results Line 1- 3
Q: 5 Can the authors somehow represent the phenotypic and treatment characteristics in a graph, figure, or flowchart? It should be clear which were the most significant information collected from this review.
R: 5 We tried to collect the most recent information in this complex malformation
- 6- Q7 The discussion section should start with the main objective of this review study and the most important results found. The collected results (subsections) by the authors should be discussed from multiple angles and placed in context without overinterpreting them. What was known, what new information did the authors collect? What does this review contribute to the current knowledge about this syndrome? A paragraph of limitations and suggestions for this review should be written before the conclusion.
6. Did the authors reach any conclusions from this review?
I would like to encourage the authors to rewrite this review, thinking about the main objective of this study, its design and responding with the results and arguments of the discussion to the most appropriate conclusion of this review.
R: 6 R:7 as correctly suggested by the reviewer we have added limitations and conclusions.
The disorder is complex in several clinical points and so many questions remain at the moment unsolved
We wish to thanks the reviewers for the appropriated and precious suggestions
Round 2
Reviewer 1 Report
The authors adequately addressed the required edits, Thank you
Author Response
Reviewer 1
Q 1. The manuscript has method and results section so it is a systematic review. Please add “A systematic review” in the title.
R 1 We have add “A systematic review” in the title.
Q 2. I have noticed few typing and grammar errors such as “Cochraine” it should be “Cochrane”, Please revise all the manuscript regarding this issue.
R2 We have ceck this issue
Q 3. The results section is very short and missing many important data. How many studies were retrieved and included in this systematic review. What was the types of the retrieved and eligible studies?. How many subjects included as the total of subjects from the included studies?. It would be better if a table describe the main characteristics of the included studies added to the manuscript.
R 3 We have added tables
Reviewer 2 Report
All comments were adequately addressed. However, I do not see the included clinical picture (figure 1).
Author Response
Reviewer 2
- 1 The manuscript is in general well structured and written, however, it is difficult to identify a major contribution to the field, despite its relation with a very rare disease. Although
data retrieved from the literature is well described, but it is not sufficiently contrasted (i.e. use of Tables to contrast clinical/genetic findings among different case series).
- 1 Presently KFS has been included in the group of disorder belonging to the PIK3CA-related overgrowth spectrum (PROS). Case-reports with chromosomal anomalies are scanty. As suggested, we reported a table I with three examples of KFS associated with chromosomal anomalies and Table II reassuming the results of Imaging features as reported by ALWALID et al.in 14 KTS patients (see Table 1 and Table 2)Line ….. Line……
- 2 In my opinion, another major limitation of this review is the absolute lack of representative clinical and radiological images of the KLIPPEL-TRENAUNAY-WEBER SYNDROME, which is of utmost importance to
provide the reader with a clear clinical picture of this disorder, which even allows the differential diagnosis.
- 2 We have included a photo of a recent child
Minor criticism
- 3 The official name of this disorder: KLIPPEL-TRENAUNAY-WEBER SYNDROME(OMIM %149000).
- 3 The official name of the disorder has been reported. Introduction Line 4
Q.4 Please review the phrase "selected from January 2003 to January 2023"
- 4 The sentence has been corrected Methods line 3
- 5 Please write the human gene symbols in italics
R 5: Italics symbol has been used in human gene
Q.6, is it refers to the absence of "curative treatment" or "treatment consensus"?
- 6 the sentence has been corrected as you suggested Management Line 1
Q The manuscript requires language editing
R The paper was review by a full professor of English of the University of Catania
Reviewer 3 Report
Even though this manuscript has improved, this improvement has not been significant. the focus of this manuscript should shift to a review based on a case report of three patients with KTS. It is suggested to review the design in detail, present the most significant points and that the best description/review of these adhere to the main objective that led them to carry out this review. If this study was supported by an Institution that manages/treats this type of patient, it should be said.
In the introduction section: ...previously mentioned... Since we have been able to follow the growth and development of various patients, for example, the KFS patient presented in 1974 as “Picture of the Month” who is now an adult (8), to a youth with reverse KFS who 12 years is currently old, and a 2-year-old boy recently diagnosed with reverse KFS. These observations led us to review the latest findings on this syndrome. Therefore, the main objective of this review was to make known the advances in genetics, the clinical characteristics, as well as the complications, the differential diagnosis and the management related to this syndrome.
Material and methods: Please specify what information from these cases (articles) was sought and collected. The description must be clear, concise, and detailed. What parameters were included and excluded? What data was essential to collect?
Results section: Describe the most important characteristics of the studies found, for example, the number of patients in total, age, gender, etc. How many patients did the authors find? How many of them reach adulthood? Male/female ratio, etc. In Table 1 what about the treatment characteristics? It should be clear which was the most significant information collected from this review.
Discussion section: If authors include data/photos of a KTS patient they should first describe this information in the M&M section. The authors must declare that this study was carried out in accordance with the ethics committee of the Institute in which they work and with the principles of the Declaration of Helsinki. In addition, it must be reported that the parents/guardians signed an informed consent.
I would like to encourage the authors to rewrite this review, thinking about the main objective of this study. When authors present a case report (three patients), they can focus the discussion generally on progress, etc., considering what was known before conducting this review.
Moderate editing of the English language is required.
Author Response
Reviewer 3
- 1 The manuscript requires language editing. The title should improve, for example “Klippel-Trenaunay syndrome, segmental/focal overgrowth malformation: a review”. The main aim must be direct and the same throughout the manuscript. References should be recent and relevant, they should be well referenced, and their use should be improved throughout the manuscript.
- 1 The text has been reviewed by an English Native reviewer. The title of the text has been changed as suggested
Q 2 The introduction section needs improvements. It would be better to write this here: (KTS). A proper presentation and a good and clear justification (reason) for conducting this comprehensive review should be given. Why was this review done?
R. 2 KTS acronym has been reported at the beginning, The reason of this comprehensive review has been added at the end of the introduction section (Last 5 lines of the introduction 9
- 3 The methods section is sparse and need deep improvements. What kind of review was it? The description must be clear, concise, and detailed. What parameters were included and excluded? What data was essential to collect?
R: 3 Method section has been modified Methods Line 6-9
Q.4 The results section should improve: How many articles did this search return? Describe the most important characteristics of the studies found, for example, the number of patients in total, age, gender, etc. Avoid repeating the same information.
R, 4 Result section has been modified Results Line 1- 3
Q: 5 Can the authors somehow represent the phenotypic and treatment characteristics in a graph, figure, or flowchart? It should be clear which were the most significant information collected from this review.
R: 5 We tried to collect the most recent information in this complex malformation
- 6- Q7 The discussion section should start with the main objective of this review study and the most important results found. The collected results (subsections) by the authors should be discussed from multiple angles and placed in context without overinterpreting them. What was known, what new information did the authors collect? What does this review contribute to the current knowledge about this syndrome? A paragraph of limitations and suggestions for this review should be written before the conclusion.
6. Did the authors reach any conclusions from this review?
I would like to encourage the authors to rewrite this review, thinking about the main objective of this study, its design and responding with the results and arguments of the discussion to the most appropriate conclusion of this review.
R: 6 R:7 as correctly suggested by the reviewer we have added limitations and conclusions.
The disorder is complex in several clinical points and so many questions remain at the moment unsolved
We wish to thanks the reviewers for the appropriated and precious suggestions .